# Multiscale Characterization of an Oxide Scale Formed on the Creep-Resistant ATI 718Plus Superalloy during High-Temperature Oxidation

**DOI:** 10.3390/ma14216327

**Published:** 2021-10-23

**Authors:** Adam Kruk, Aleksander Gil, Sebastian Lech, Grzegorz Cempura, Alina Agüero, Aleksandra Czyrska-Filemonowicz

**Affiliations:** 1Faculty of Metals Engineering and Industrial Computer Science, International Centre of Electron Microscopy for Materials Science, AGH University of Science and Technology (AGH-UST), al. Mickiewicza 30, 30-059 Krakow, Poland; kruczek@agh.edu.pl (A.K.); cempura@agh.edu.pl (G.C.); czyrska@agh.edu.pl (A.C.-F.); 2Faculty of Materials Science and Ceramics, AGH University of Science and Technology (AGH-UST), al. Mickiewicza 30, 30-059 Krakow, Poland; gil@agh.edu.pl; 3Departamento de Materiales y Estructuras, Instituto Nacional de Técnica Aeroespacial (INTA), Carretera de Ajalvir, Km 4, 28850 Torrejón de Ardoz, Spain; agueroba@inta.es

**Keywords:** creep-resistant, superalloys, 718Plus, SEM, TEM, STEM, FIB-SEM tomography, high-temperature corrosion, internal oxidation

## Abstract

The ATI 718Plus^®^ is a creep-resistant nickel-based superalloy exhibiting high strength and excellent oxidation resistance in high temperatures. The present study is focused on multiscale 2D and 3D characterization (morphological and chemical) of the scale and the layer beneath formed on the ATI 718Plus superalloy during oxidation at 850 °C up to 4000 h in dry and wet air. The oxidized samples were characterized using various microscopic methods (SEM, TEM and STEM), energy-dispersive X-ray spectroscopy and electron diffraction. The 3D visualization of the microstructural features was achieved by means of FIB-SEM tomography. When oxidized in dry air, the ATI 718Plus develops a protective, dense Cr_2_O_3_ scale with a dual-layered structure. The outer Cr_2_O_3_ layer is composed of coarser grains with a columnar shape, while the inner one features fine, equiaxed grains. The Cr_2_O_3_ scale formed in wet air is single-layered and features very fine grains. The article discusses the difference between the structure, chemistry and three-dimensional phase distribution of the oxide scales and near-surface areas developed in the two environments. Electron microscopy/spectroscopy findings combined with the three-dimensional reconstruction of the microstructure provide original insight into the role of the oxidation environment on the structure of the ATI 718Plus at the nanoscale.

## 1. Introduction

Water vapor is not found solely in the Earth’s atmosphere; it is also a constituent of many gaseous environments closely associated with human activity. It results from the evaporation of rain and water masses, and it is a product of the combustion of the hydrocarbons found in solid, liquid, and gaseous fossil fuels. The conversion of the chemical energy stored in such fuels into thermal energy is therefore always accompanied by the emission of fumes rich in water vapor, which is particularly true with regard to the conventional power industry as well as all forms of transport—land, air or marine. Since hydrocarbon combustion occurs at high temperatures, the influence of water vapor on gas-induced corrosion of various kinds of metallic materials (e.g., creep-resistant steels and nickel-based superalloys) has now been studied intensively for several decades. The impact of water vapor on corrosion processes is complex. It is occasionally observed that the adhesion of the Cr_2_O_3_ scale, which is formed on corrosion-resistant steels and nickel-based superalloys, to the alloy substrate improves in the presence of water vapor despite its increased growth rate [1,2,3,4,5,6]. This is caused by a change in the mechanism that underlies the transport of reagents in this scale—when the inward diffusion of oxygen is predominant, a fine-grained Cr_2_O_3_ scale with excellent adhesion to the bulk metallic material forms. However, water vapor is also responsible for a number of undesirable effects that accelerate the corrosion-induced deterioration of metals and alloys. These effects are mostly associated with the formation of volatile chromium oxyhydroxides, for example, CrO_2_(OH)_2_ [7,8]. The presence of such compounds causes the protective properties of the Cr_2_O_3_ scale to deteriorate, reducing the maximum operating temperature and useful life of the alloys. Corrosion in pure steam, which occurs on the inner surface of power boiler waterwalls and in gas turbines, among others, is a particularly notable case. The role of water vapor in the high-temperature corrosion of metal alloys was described in detail in the literature on the subject [9].

The ATI 718Plus^®^ (hereafter referred to as the 718Plus), an advanced creep-resistant polycrystalline nickel-based superalloy, has high strength and corrosion resistance that are both superior to those of the well-known, conventionally applied Inconel 718 (hereafter the IN718) while still offering excellent weldability and processing characteristics. With a maximum working temperature of about 700 °C, the 718Plus is suitable for the construction of static and rotating critical components in aircraft as well as power generation systems [10,11]. The oxidation resistance of this superalloy was previously investigated in the air at temperatures in the range of 650–1000 °C [12,13,14,15,16,17,18,19,20,21]. The results showed that the material develops a protective Cr_2_O_3_ scale during oxidation. Beneath the scale, an internal oxidation zone with a Cr-depleted matrix and alumina precipitates forms. In our previous study [12], we showed that a thin layer of the δ-Ni_3_Nb phase (a so-called interlayer) also forms beneath the Cr_2_O_3_ scale.

Although these findings were significant, the oxidation tests performed in the above-mentioned studies were conducted mainly in dry air. There is limited literature data on the oxidation of the 718Plus in more aggressive environments, i.e., those containing water vapor. B.A. Pint et al. [17,18] investigated the oxidation of the IN718 and its derivatives in various extreme environments in order to demonstrate the role of water vapor in depleting Cr in the presence of oxygen. They reported that after oxidation at 650 and 750 °C, there is a considerable difference in the mass gains observed for ambient laboratory air and wet air with 10 vol.% of H_2_O. At higher temperatures, the Cr depletion associated with the presence of water vapor accelerates even further. Furthermore, oxidation at 750 and 800 °C is increasingly internal in character as Al, Ti, and Nb start to oxidize internally.

The present study is focused on the morphological and chemical characterization of the 718Plus superalloy after oxidation at 850 °C up to 4000 h in two different media—wet air (10 vol.% of H_2_O) and dry air, as well as the comparison of the results. One of the main objectives was to describe the differences between the structure, chemical and three-dimensional phase distribution of oxide scales formed on the 718Plus and its near-surface areas, as developed in the two investigated environments. The present study is a continuation of our earlier research on the effect of long-term thermal exposure as well as high temperature oxidation in dry air on the microstructure of ATI 718Plus superalloy. The results of these investigations were published in Refs [14,21,22].

## 2. Material and Methods

### 2.1. Material

The ATI 718Plus^®^ superalloy (ATI Specialty Materials, Pittsburgh, PA, USA) with the nominal chemical composition Ni-18Cr-9.7Fe-9.2Co-5.5(Nb + Ta)-2.7Mo-1W-1.5Al-0.7Ti-0.02C (wt.%) is produced by the manufacturer using vacuum induction melting (VIM) and vacuum arc re-melting (VAR) followed by rotary forging and ring rolling. The alloy undergoes a standard heat treatment: solution treatment at 968 °C (1 h, oil quenching) followed by two-step ageing at 788 °C (8 h, furnace-cooled) and 704 °C (8h, air-cooled). The resulting microstructure consists of a γ matrix (Ni-based solid solution) strengthened with spherical γ’-Ni_3_(Al, Ti, Nb) precipitates and plate-like η-Ni_6_AlNb precipitates formed mainly at grain boundaries [19,20,21,22,23]. Some other minor phases (primary carbonitrides, δ-Ni_3_Nb, η-Ni_3_Ti, and Ni_6_(Al,Ti)Nb particles and/or TCP phases) were also observed in the 718Plus depending on its exact chemical composition and heat treatment [19,23,24].

The investigated 718Plus superalloy was received in the form of 20 mm bars; its microstructure is described in detail in Refs [14,20,21].

Oxidation studies were conducted at 850 °C, using two different gas atmospheres: dry air, and humidified air with a 10 vol.% H_2_O content. This temperature is higher than the typical operating temperature of the 718Plus and was chosen to accelerate the oxidation processes and the microstructural changes and thus simulate the long-term operation of this alloy at lower temperatures [18]. Oxidation took place under quasi-isothermal conditions, which means that the samples were removed from the furnace after certain time intervals and were weighed after they had cooled to ambient temperature. A series of four samples (20 mm × 10 mm × 5 mm) was prepared for each of the two atmospheres—one for every tested oxidation time, i.e., for 120, 1000, 2000 and 4000 h. The specimens were ground with 180 grit and cleaned in acetone prior to exposure. The description of the rig is provided elsewhere [25]. Dry air was bubbled through deionized water kept at 45 °C in order to obtain the 10 vol.% H_2_O air atmosphere.

The oxidized samples (labelled “dry” and “wet” depending on the oxidation medium) were characterized using the methods described below.

### 2.2. Characterization Methods

#### 2.2.1. X-ray Diffractometry and Electron Microscopy/Spectroscopy

Phase analysis of the oxidized samples was performed by means of X-ray diffractometry (XRD) using a D500 Kristalloflex X-ray diffractometer (Siemens, Monachium, Germany) operating under monochromatic Cu-K_α1_ (1.54 Å) radiation. The Bragg–Brentano (BB) geometry was applied.

The phases formed in the oxide scale and the near-surface area were characterized using scanning, transmission and scanning-transmission electron microscopy (SEM, TEM and STEM, respectively), energy-dispersive X-ray spectroscopy (EDXS) and electron diffraction. Details about sample preparation for SEM (sample cross-section with silica finish) and TEM/STEM (FIB lamellae) are given in Ref. [22].

The SEM observations were conducted using a high-resolution Merlin Gemini II microscope (ZEISS, Oberkochen, Germany) equipped with a Schottky field-emission gun (Bruker, Billerica, MA, USA) and an EDX detector with a Quantax 800 microanalysis system (Bruker, Billerica, MA, USA). The SEM images were obtained using secondary electrons (SE) and backscattered electrons (BSE).

The TEM/STEM observations were performed using a Tecnai G2 20 Twin microscope (FEI, Waltham, MA, USA) and a probe-corrected Titan Cubed G2 60–300 microscope (FEI, Waltham, USA) equipped with the ChemiSTEM™ system [26]. The STEM imaging with high-angle annular-dark-field (HAADF) contrast and EDXS mapping was used to characterize the phases’ nanostructure down to the atomic level. Phase identification was conducted using selected area electron diffraction (SAED) combined with HRSTEM and EDXS microanalysis. The electron diffraction patterns and high-resolution STEM (HRSTEM) images were interpreted with the help of the JEMS software (v4.92) [27]. The STEM-EDXS data were acquired at 300 kV and then underwent analysis using the Esprit software (Bruker, v1.9), in which the standardless Cliff-Lorimer quantification method was applied.

#### 2.2.2. Atomic Force Microscopy and FIB-SEM Tomography

The morphology and topography of the oxide scales formed on the 718Plus superalloy were investigated using a commercial CoreAFM (Nanosurf, Liestal, Switzerland) atomic force microscope (AFM) operating in the dynamic force mode (tapping mode). For measurements, a probe with an 8 nm tip radius (HQ: NSC16/AL BS; MikroMasch, Sofia, Bulgaria) as well as a cantilever with low-resonance frequency (190 kHz) and a 45 N/m force constant was used. To collect the data, 20 µm × 20 µm areas were scanned and further analyzed using the Gwyddion software (v2.56).

The 3D characterization of the oxide scale and the near-surface area of the samples oxidized for 120 h was carried out by means of FIB-SEM tomography. Data for tomographic reconstruction was collected using the slice-and-view technique and by employing the NEON CrossBeam 40EsB (ZEISS) microscope. Details about the setup and procedure used for data acquisition are given in Ref. [25]. The SEM images were obtained at an accelerating voltage of 2 kV as well as with both SE and BSE detection. The serial FIB slicing (approximately 24 nm steps) was performed using a Ga^+^ ion beam at a voltage of 30 kV and a beam current of 1 nA. Greyscale images with a resolution of 1024 × 768 pixels were obtained. For the “dry” sample, 269 images with a voxel size of 18 nm × 18 nm × 36 nm were obtained, while for the “wet” sample, the number of obtained images was 392 and their voxel size was 12 nm × 12 nm × 24 nm. Altogether, raw data were obtained for a total sample volume of about 2437 µm^3^ (for the “dry” sample) and 1055 µm^3^ (for the “wet” sample). Datasets were processed, reconstructed and visualized in 3D using the Fiji and Avizo 6.3 applications. Basic processing included stack alignment, image processing and segmentation based on the Z-contrast. The FIB-SEM tomographic reconstructions were cropped to 10.8 µm × 8.4 µm × 9 µm cuboids to allow them to be compared using the same scale.

## 3. Results and Discussion

Figure 1 presents curves that show the mass change of 718Plus superalloy samples oxidized in dry and wet air. For each type of atmosphere, the values for measurement points up to 120 h are mean values measured for four samples, while those for measurement points between 120 h and up to and including 1000 h are mean values for three samples. The values for measurement point between 1000 h and up to and including 2000 h are mean values measured for two samples. Finally, the values for oxidation times above 2000 h were obtained for a single sample oxidized for a total time of 4000 h. The authors are aware that the interpretation of quasi-isothermal mass change curves is prone to error, since every time the sample is cooled down to ambient temperature, its subsequent corrosion process may be affected. The conclusions drawn based on the obtained mass change curves were therefore very general. It was found that up to approximately 1500 h, the mass gain recorded for the 718Plus sample oxidized in wet air was higher than for the sample oxidized in dry air by approximately 20%. For longer oxidation times under the wet air atmosphere, mass loss was observed. It is presumed that this effect was associated with the formation of volatile chromium oxyhydroxides—mostly CrO_2_(OH)_2_—as a result of the reaction between Cr_2_O_3_ and O_2_ as well as H_2_O. It should, however, be noted that when oxidation takes place under quasi-isothermal conditions, the observed mass loss may also be due to the oxide scale spallation caused by stresses that are generated in the oxide-metal system as the sample is cooling down to ambient temperature. The co-occurrence of these two effects cannot be ruled out in this case. For the sample oxidized in dry air, on the other hand, continuous mass gain was observed; however, spallation of the oxide scale is also possible.

Figure 2 presents the XRD spectra recorded in the B-B geometry for the 718Plus superalloy samples oxidized for 4000 h in dry and wet air. The samples did not differ in terms of phase composition. The analysis of the spectra revealed that the main oxide phase in the oxidation product was Cr_2_O_3_. Intense peaks originating from the γ and δ-Ni_3_Nb phases were also identified.

Figure 3 shows the surface morphology of the scales formed on the 718Plus superalloy in dry and wet air after oxidation of 120, 1000 and 4000 h. These observations show that the scales’ surface morphology differs to a significant degree. After 120 h of oxidation in dry air, the surface of the corresponding scale features numerous plate-like Cr_2_O_3_ crystallites spaced apart from one another. The crystallites were approximately 3–4 μm in length, while their thickness did not exceed 1 μm. They protruded from the surface of an inner layer composed of fine grains with diameters of approximately several dozen hundred nanometres. As oxidation times extended, the crystallites increased in size, lost their plate-like shape, and transformed into a compact layer, which is most evident in case of the sample oxidized for 4000 h. During oxidation in wet air, a fine-grained scale was formed on the 718Plus superalloy; no large, plate-like Cr_2_O_3_ crystallites similar to those observed for the samples oxidized in dry air were found in this case. After 120 h of oxidation in wet air, the size of crystallites found on the surface of the scale did not exceed 1 μm, and chemical composition analysis showed that these crystallites contained titanium (as shown later in Figure 6). For longer oxidation duration, the grains grew larger, but this process was very gradual, and only a few of them were more than 3 μm in size after 4000 h of oxidation.

The results of the SEM analyses described above showed that there are significant differences in scales’ surface morphology and size of the oxide crystallites depending on the oxidation environments. 

Atomic force microscopy was used to study the oxidation products formed on the 718Plus superalloy during oxidation in dry and wet air at 850 °C. The AFM made it possible to obtain detailed topographic information of the real surface of the samples oxidized in dry and wet air. Figure 4 shows typical images of the surface topography of the samples oxidized for 120 h in dry (left) and wet (right) air. The images show that the chromia scales formed during oxidation in either environment were crystalline in character. 

The results of quantitative AFM evaluations are presented in Table 1. The following parameters were measured:

S_q_ [nm]—RMS roughness: the root mean square (RMS) of the height of all points in the 3D scan,

S_t_ [μm]—the maximum peak-to-valley height for the entire image,

SAF (surface area factor)—the ratio of the real 3D surface area to that of its 2D projection in a direction normal to the average sample surface.

The increased values of the S_q_ and SAF parameters indicate that both surface layers exhibited pronounced development. The oxide crystallites formed on the surface of the specimen oxidized in dry air are slightly taller (higher S_q_ and S_t_ values) than those formed in wet air after the same oxidation time. These results are consistent with the results of the SEM images of the sample surfaces oxidized in the two environments (as shown in Figure 3). 

Figure 5 compares the microstructures of oxide scales formed on the 718Plus superalloy after oxidation in dry (A) and wet (B) air. These TEM observations were conducted using lamellae prepared via FIB. They are consistent with the previously presented results of SEM observations, and they confirm that there are significant differences in scale microstructure depending on the oxidation atmosphere. The scale formed in dry air has two layers: a coarse-crystalline outer layer (white arrows—Figure 5A) and a fine-crystalline inner layer. On the other hand, while the scale formed after oxidation in wet air has a similar thickness; it consists only of a single layer and is composed predominantly of fine, equiaxed grains with a size of up to 200 nm and only a few slightly larger grains on the outer surface (white arrows—Figure 5B), as in case of the inner layer of the scale formed in dry air.

It is widely accepted that the growth of grains in the form of columnar polyhedrons with flat sides and sharp edges is a consequence of the outward diffusion of a metal across point defects in the cation sublattice of a given oxide. On the other hand, in scales composed of fine and equiaxed grains, the diffusion of oxygen across numerous grain boundaries may be the predominant mechanism via which mass is transported within the oxide layer. Such a mechanism cannot be ruled out in the case of the Cr_2_O_3_ scale formed in wet air, and the same is true of the inner layer of the Cr_2_O_3_ scale formed in dry air. However, Latu-Romain et al. [28], who oxidized chromium at an oxygen partial pressure of 10^−12^ atm. and a temperature of 900 °C, reported that the inner layer in a dual-layered Cr_2_O_3_ scale composed of fine, equiaxed grains grew as a result of the inward diffusion of oxygen across point defects in the anion sublattice of this oxide. Both inward diffusion mechanisms involving oxygen, namely, diffusion across anion vacancies and diffusion across grain boundaries, should be considered viable in the case of the investigated samples, and it is impossible to determine which of the two mechanisms is predominant based on microscopic observations alone.

Figure 6 presents the STEM-HAADF images of the microstructures of the 718Plus samples oxidized in dry and wet air, as well as the maps of the distribution of selected elements in the scales and the region beneath the scales. It can be concluded from these maps that in case of both atmospheres, the selective oxidation of chromium led to similar changes in the microstructure and phase composition of the superalloy in the immediate vicinity of the oxide layer. Previous research by the authors [12] showed that a layer composed of the δ-Ni_3_Nb phase had a tendency to form in the internal oxidation zone characterized by chromium depletion.

Electron diffraction (SAED) analyses of the 718Plus microstructure confirmed the presence of numerous precipitates of the δ phase between the scale and the superalloy matrix (area 2 in Figure 6) and the presence of γ-Al_2_O_3_ particles in the internal oxidation zone (area 3 in Figure 6). The SAED analysis of the area 1 of the scale formed in wet air revealed the presence of TiO_2_. Numerous precipitates of this phase were also found in the inner layer of the Cr_2_O_3_ scale formed in dry air, and in the region of the superalloy just beneath the oxide layer—in case of both oxidation atmospheres. It is thought that the growth of TiO_2_ crystallites on the surface of the Cr_2_O_3_ scale occurs as a result of the outward diffusion of titanium across grain boundaries in chromia [29], whereas its precipitation in the scale interior and the bulk of the alloy is associated with the internal oxidation of this element.

Measurements of oxidation mass gain and the thickness of the scales, performed using polished specimens, indicate that the oxide layer formed in wet air is slightly thicker than its counterpart formed in dry air. The difference is not pronounced enough, however, to result in significant differences in the structure of the layer composed of the δ phase in the zone beneath the scale. This layer is discontinuous, has similar thickness, and consists of precipitates of similar size for both types of samples (Figure 7).

Figure 7 shows a 3D visualization of the tomographically reconstructed microstructure of the scales formed on the 718Plus superalloy after 120 h of oxidation in dry (Figure 7A,C) and wet (Figure 7B,D) air at 850 °C and the areas beneath them. The images reveal microstructural features of individual components: two oxide phases (Cr_2_O_3_ and Al_2_O_3_) and two intermetallic phases (δ and η) marked using different colors. This 3D visualization did not reveal significant differences in the morphology and phase composition of the layers obtained on the samples oxidized in dry air and those oxidized in wet air. In both cases, a discontinuous layer consisting of the δ phase was found at the scale/superalloy interface, and in both cases this layer had a well-developed surface (Figure 7C,D). The morphology of this layer differs to a significant degree from the morphology of the δ phase precipitates observed for the IN718 and the 718Plus superalloys, which have a plate-like character. The occurrence of open porosity in the δ phase layer (Figure 7C,D) facilitates diffusion both for oxygen, which uses this path to travel to the superalloy material and form an internal oxidation zone, and for aluminum, which travels in an outward direction—to the scale layer, in which the presence of Al_2_O_3_ is also observed. It can be stipulated that the formation of this discontinuous layer may to a certain extent inhibit the inward diffusion of oxygen and the outward diffusion of Cr and Al, which may be an important factor that reduces the growth rate of the scale and the internal oxidation zone.

Figure 8 presents the results of microscopic analyses conducted at resolutions down to the atomic level. Microscopic analyses are supplemented with STEM-EDXS distribution maps of selected chemical elements. Figure 8A shows a STEM-HAADF image of the scale and the area underneath, highlighting the presence of δ phase particles in this area (Figure 8B). The HRSTEM-HAADF imaging revealed columns of δ-Ni_3_Nb phase atoms (Figure 8C). A JEMS-simulated HAADF image of the same δ-Ni_3_Nb arrangement is superimposed onto the image in this figure (highlighted in blue), while the fast Fourier transform (FFT) of this image is shown in the rightmost part of this figure. After comparing the symmetry and spot distances in the FFT image with the diffractograms of the δ-Ni_3_Nb phase calculated using JEMS for various orientations (zone axis, ZA), it can be concluded that the high-resolution HAADF image corresponds to [−110] zone axis of the δ-Ni_3_Nb phase. The HAADF image calculated for this orientation matched the experimental high-resolution image (Figure 8C). In consequence, interplanar distances (d_h,k,l_) of the δ-Ni_3_Nb phase marked in Figure 8C allowed this phase to be identified based on the crystallographic data from the ICSD database. It should be noted that when detecting high-angle scattering, the intensity reaching the HAADF detector is almost proportional to the square of the atomic number (Z^2^), allowing the strong chemical contrast (Z-contrast [30]) of atom columns in their actual positions to be determined. The bright and dark spots observed in the δ-Ni_3_Nb phase therefore correspond to Nb and Ni, respectively. Figure 8D shows superimposed STEM-EDXS maps of selected chemical elements, and these maps confirm the chemical and crystallographic complexity of the particle shown in Figure 8B. Figure 8E presents FFT images for the γ and δ-Ni_3_Nb phases as well as their superimposition, from which it follows that the g_γ_ = [1−1−1] and g_δ_ = [400] reflections are very close, indicating that the interplanar distance between the {111}_γ_ and {004}_δ_ crystallographic planes is small. These planes are also parallel to one another. Figure 8F shows the HRSTEM-HAADF image of the δ-Ni_3_Nb precipitate and the γ matrix atom columns. The rightmost small figures present a HAADF image of the γ phase nanostructure and its image simulated by JEMS (highlighted in blue) as well as an FFT for this phase. The presented analyses lead to the conclusion that δ-Ni_3_Nb particles precipitate from a supersaturated γ matrix and that the (111)_γ_‖(004)_δ_ crystallographic orientation is preserved. This relationship was also found in the literature [31,32].

Figure 9 presents a bar graph that shows the thickness of the different scales and the depth of internal oxidation zones in the 718Plus superalloy after oxidation in dry (left-hand side) and wet air (right-hand side). The graph was plotted based on SEM images of polished cross-sections taken in five randomly selected locations. Each depth of the internal oxidation zone was calculated as an arithmetic mean of the deepest and most shallow locations of Al_2_O_3_ precipitates found in the area delimited using the dotted line in Figure 10. The graph shows that a slightly thicker scale forms on the 718Plus superalloy when it is oxidized in wet air; in addition, the internal oxidation zone is deeper in case of such sample. 

Zurek and co-workers [32] observed that breakdown of protective chromia scales in water vapor-rich atmospheres is correlated with an increase in internal oxidation as a result of hydrogen dissolution in an Fe-10Cr model alloy. Indeed, Essuman et al. [33] proposed that hydrogen, which is dissociated from water vapor, promotes internal oxidation of aluminum by increasing the solubility and/or diffusivity of oxygen in Al containing Fe-Cr alloys. Internal oxidation of Cr in Fe–Cr alloys is accelerated. 

To summarize, the present study is focused on multiscale 2D and 3D characterization (morphological and chemical) of the scale formed on ATI 718Plus superalloy during high-temperature oxidation (850 °C) up to 4000 h in dry and wet air. The state-of-the-art microscopic methods (SEM, TEM/HRTEM and STEM/HRSTEM) down to atomic level, energy-dispersive X-ray spectroscopy and electron diffraction were applied to characterize the oxidized samples. The 3D visualization of the oxide scale and beneath was achieved utilizing FIB-SEM tomography. 

The results of these analyses showed significant differences in scales’ surface morphology and size of the oxide crystallites depending on the oxidation environments. The Cr_2_O_3_ scale formed in dry air was composed of two layers. The outer layer consisted of large crystallites, whereas the inner layer was fine-crystalline. The oxide scale formed in wet air was single-layered and consisted of fine-grained crystallites; some formed on the surface of the Cr_2_O_3_ scale containing titanium was identified as TiO_2_. Numerous precipitates of this phase were also found in the inner layer of the Cr_2_O_3_ scale formed in dry air and the region of the superalloy just beneath the oxide layer, in the case of both oxidation atmospheres. It is thought that the growth of TiO_2_ crystallites on the surface of the Cr_2_O_3_ scale occurs due to the outward diffusion of titanium across grain boundaries in chromia. Furthermore, its precipitation in the scale interior and the bulk of the alloy is associated with the internal oxidation of this element. 

The scale formed in wet air was slightly thicker than the one formed in dry air, and was clearly visible after a longer period of oxidation. Nevertheless, the differences in scale thickness were not significant enough for chromium depletion in the bulk material immediately beneath the scale, causing noticeable differences in the size and structure of the δ-Ni_3_Nb phase interlayer. It should be noted that the δ-Ni_3_Nb phase was formed directly underneath the Cr_2_O_3_ scale and formed a discontinuous layer (so-called interlayer) between the chromia scale and the bulk metallic material. The morphology of this layer differs to a significant degree from the morphology of the δ phase precipitates observed for the IN718 and the 718Plus superalloys, which have a plate-like character. The occurrence of open porosity in the δ phase layer facilitates diffusion both for oxygen, which uses this path to travel to the bulk material and form an internal oxidation zone, and for aluminum, which travels in an outward direction, to the scale layer, where the presence of Al_2_O_3_ is also observed. The microstructure of the underlying bulk superalloy changed after exposure at 850 °C in comparison to that in the as-received alloy, and growth of γ’ and η phases was observed. The bulk material microstructure was investigated in detail in our earlier study. The results of these investigations are presented in Refs [14,21,22].

Electron microscopy/spectroscopy findings combined with the three-dimensional reconstruction of the superalloy microstructure provide a description of the alloy micro/nanostructure in detail and essential information concerning the morphology, composition, and 3D distribution of phases formed during oxidation in various environments in alloy’s near-surface area. These findings supply new original insight into the role of high-temperature oxidation, influencing superalloy application and service, on the structure of the ATI 718Plus down to the nanoscale. 

## 4. Conclusions

The major conclusions, based on the results of oxidation studies conducted for the ATI 718Plus superalloy using dry and wet air as the reaction media, are as follows:

1. The Cr_2_O_3_ scale formed in dry air was composed of two layers. The outer layer consisted of large crystallites, whereas the inner layer was fine-crystalline. The Cr_2_O_3_ scale formed in wet air consisted of a single fine-grained layer.

2. The scale formed in wet air was slightly thicker than the one formed in dry air and clearly visible after a longer period of oxidation. The differences in scale thickness were nevertheless not significant enough for chromium depletion in the superalloy layer immediately beneath the scale to cause noticeable differences in the size and structure of the area where the δ-Ni_3_Nb phase could be found. 

3. A discontinuous δ-Ni_3_Nb interlayer was found directly underneath the Cr_2_O_3_ scale. The morphology of this phase significantly differs from the plate-like morphology of the δ phase precipitated in the 718Plus bulk superalloys.

4. The internal oxidation zone developed in wet air, containing the Al_2_O_3_ phase, reached deeper into the bulk material by approximately 30%.

5. The HRSTEM-HAADF imaging revealed atom columns of the γ matrix and δ-Ni_3_Nb particles. It clearly confirms the precipitation of the δ-Ni_3_Nb phase during the oxidation as an interlayer formed between the chromia scale and the bulk metallic material.

6. Electron microscopy/spectroscopy findings combined with a 3D reconstruction of the micro/nanostructure provided novel insight into the role of the oxidation environment on the complexity of the oxidized 718Plus structure, in which several phases are present.

## Figures and Tables

**Figure 1 materials-14-06327-f001:**
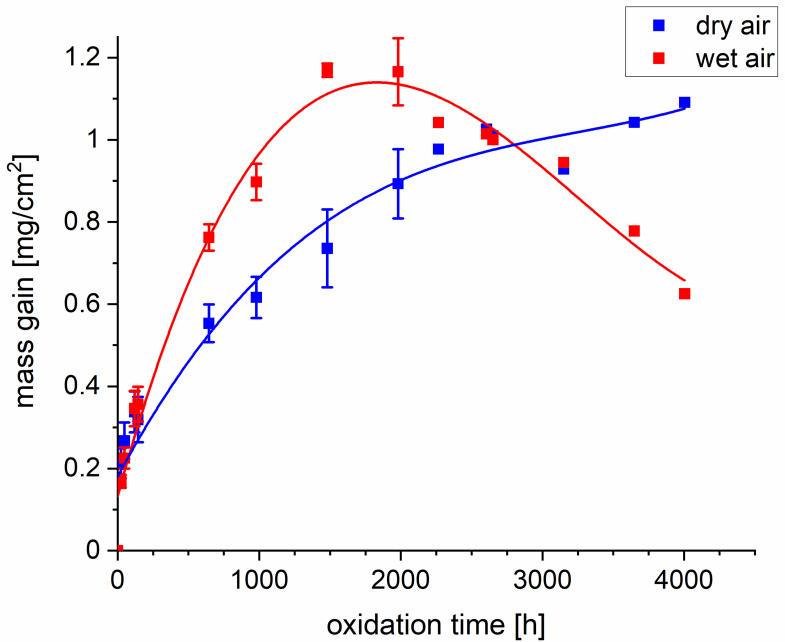
Mass gain recorded for the 718Plus superalloy oxidized at 850 °C up to 4000 h in dry and wet air.

**Figure 2 materials-14-06327-f002:**
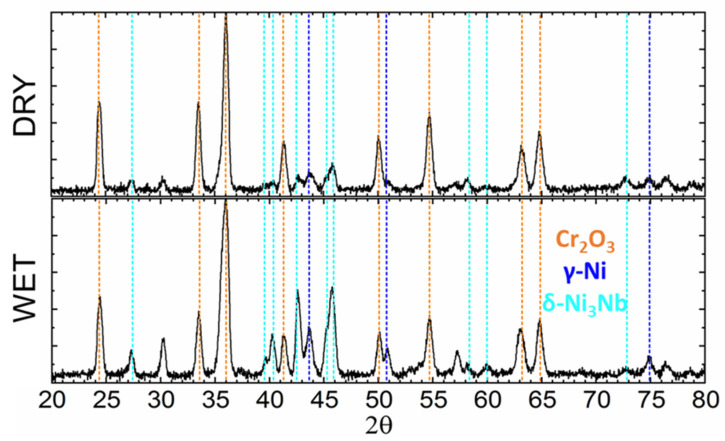
X-ray diffraction patterns obtained for the oxide scale formed on the 718Plus superalloy after oxidation at 850 °C for 4000 h in dry and wet air; XRD Bragg–Brentano geometry.

**Figure 3 materials-14-06327-f003:**
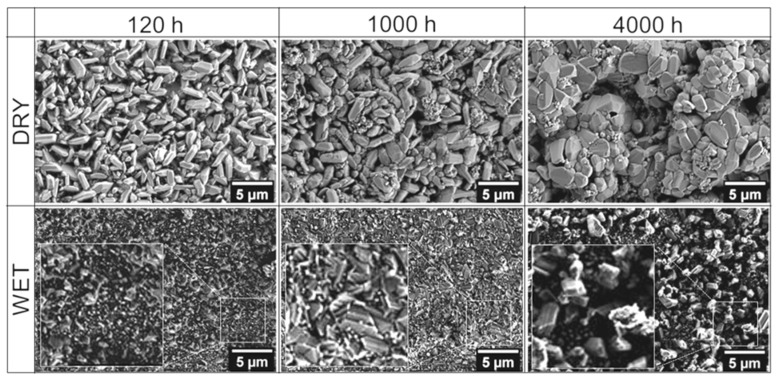
Morphology of the oxide scale on the 718Plus superalloy after oxidation at 850 °C for 120, 1000 and 4000 h in dry and wet air; SEM-SE images.

**Figure 4 materials-14-06327-f004:**
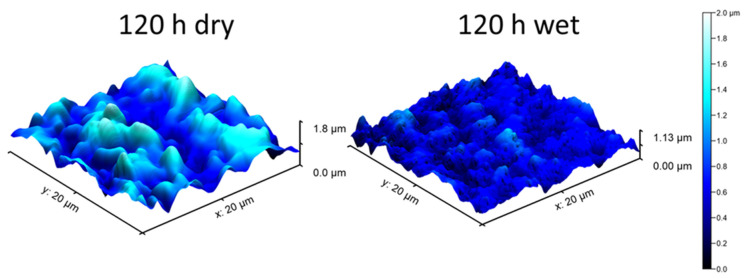
Surface topography of the 718Plus superalloy after oxidation at 850 °C for 120 h of in dry and wet air; AFM.

**Figure 5 materials-14-06327-f005:**
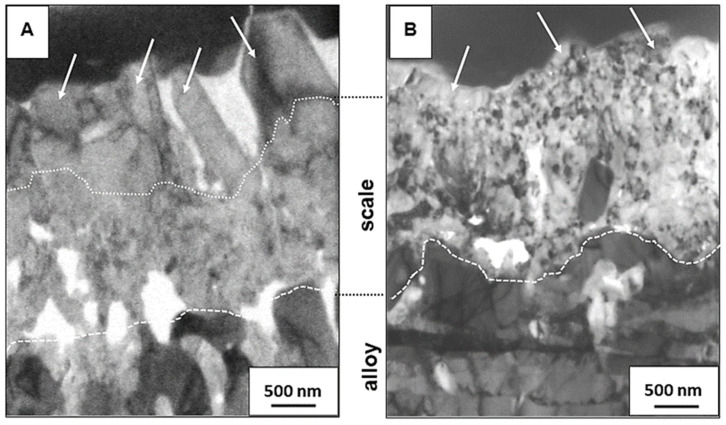
Through-scale microstructure of the 718Plus superalloy after oxidation at 850 °C for 120 h in dry (**A**) and wet (**B**) air; TEM-BF images.

**Figure 6 materials-14-06327-f006:**
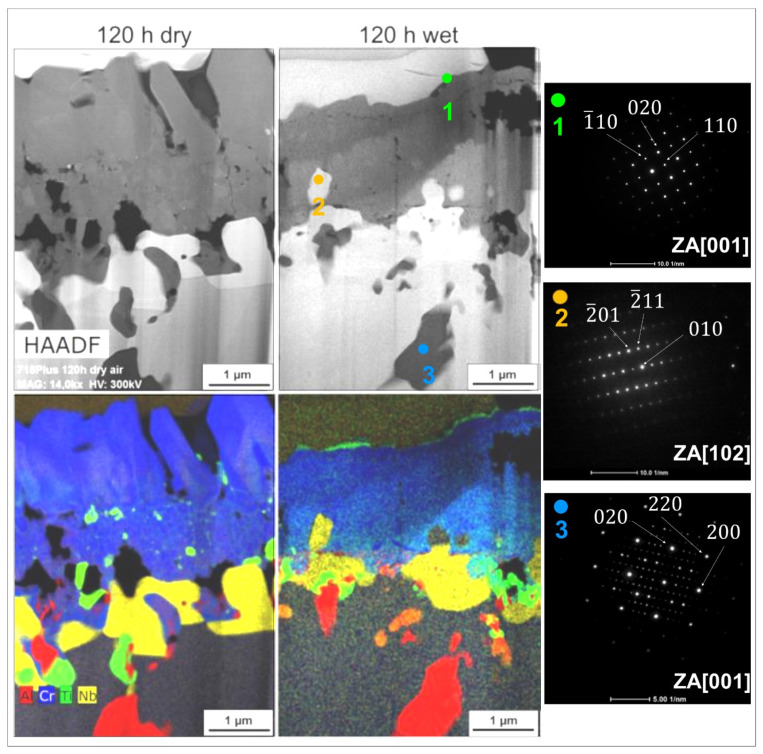
The 718Plus sample cross-sections after oxidation at 850 °C for 120 h in dry (left) and wet (right) air: microstructure (STEM-HAADF) and corresponding selected area diffraction patterns taken from areas 1–3 marked in the microstructure images as well as chemical maps of selected elements (STEM-EDXS).

**Figure 7 materials-14-06327-f007:**
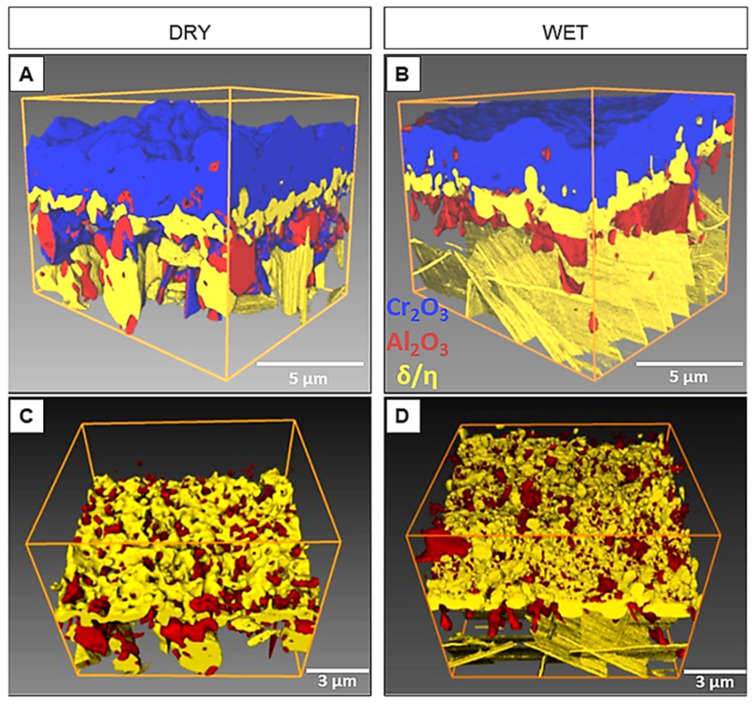
The 3D visualization of the oxide scale formed on the 718Plus superalloy after oxidation at 850 °C for 120 h in dry (**A**) and wet (**B**) air, rendered via tomographic reconstruction; microstructural features of individual components: Al_2_O_3_, δ and η phases; (**C**)—dry air, (**D**)—wet air.

**Figure 8 materials-14-06327-f008:**
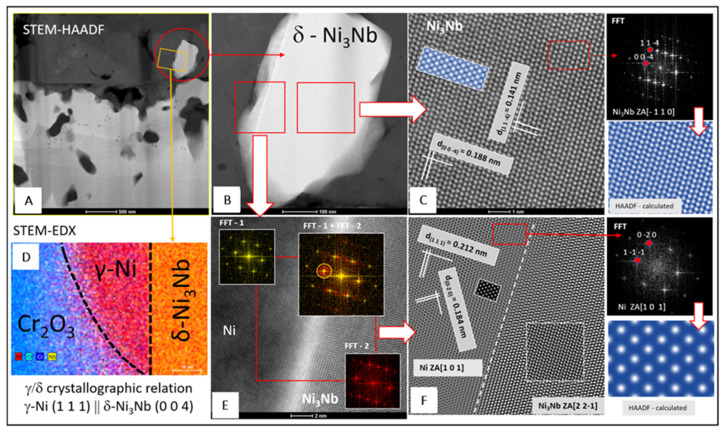
High-resolution analysis of δ phase observed in the scale layer formed on the 718Plus superalloy after oxidation at 850 °C for 120 h in wet air: (**A**) STEM-HAADF image of the scale and beneath highlighting the presence of δ-Ni_3_Nb particle particles in this area Figure (**B**), (**C**) HRSTEM-HAADF image of δ phase particle with FFT image and HAADF image simulated using JEMS software, (**D**) superimposed STEM-EDX maps of selected elements, (**E**) FFT analysis of selected areas of HRSTEM-HAADF image, and (**F**) HRSTEM-HAADF image after filtering using an FFT showing the δ-Ni_3_Nb/γ interface.

**Figure 9 materials-14-06327-f009:**
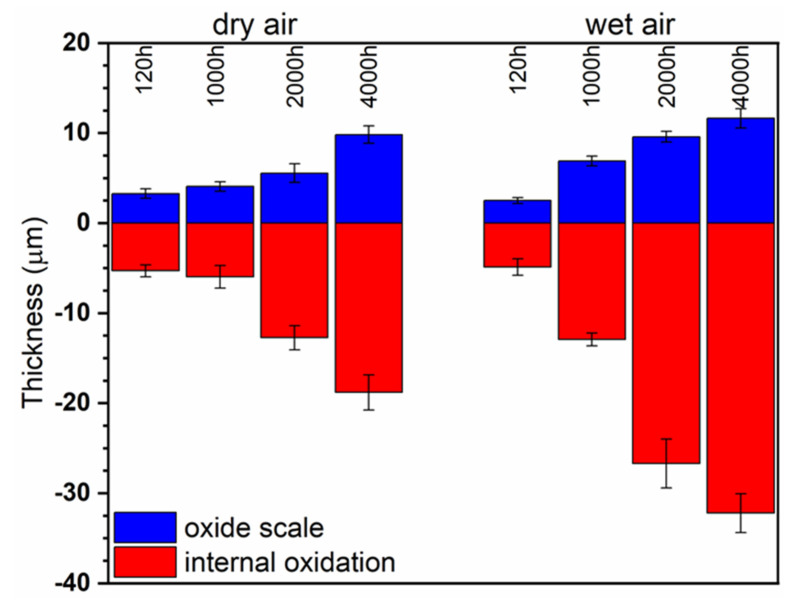
Thickness of the oxide scale and depth of internal oxidation zone formed in the 718Plus superalloy after oxidation at 850 °C for 120, 1000, 2000 and 4000 h in dry air (left) and wet (right) air.

**Figure 10 materials-14-06327-f010:**
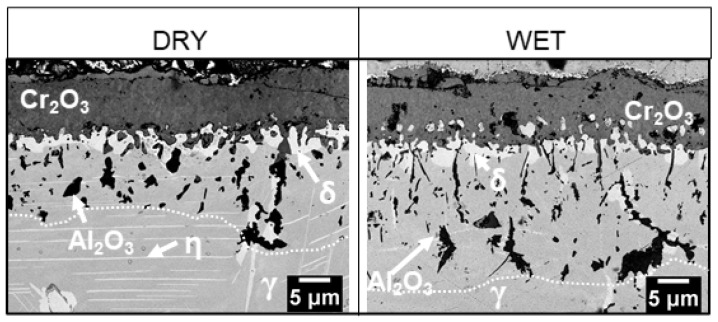
Cross-sectional microstructure of the 718Plus superalloy after oxidation at 850 °C for 4000 h in dry (left) and wet (right) air; SEM-BSE images.

**Table 1 materials-14-06327-t001:** Surface parameters (S_q_, S_t_ and SAF) determined for the 718Plus superalloy oxidized for 120 h in dry and wet air; AFM.

Surface Parameters	DRY Sample	WET Sample
S_q_ [nm]	306.5	167.8
S_t_ [μm]	1.77	1.13
SAF	1.11	1.10

## Data Availability

The raw/processed data required to reproduce these findings cannot be shared at this time as the data also form part of an ongoing study.

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
