# Peer review of "Multiscale Characterization of an Oxide Scale Formed on the Creep-Resistant ATI 718Plus Superalloy during High-Temperature Oxidation"

_materials, 2021, doi:10.3390/ma14216327_

Round 1
Reviewer 1 Report
Authors study oxidation scales in the ATI 718Plus superalloy during oxidation at 850 OC up to 4 000 hours in dry and wet air. This is an impotent question for superalloys,Thus, the paper was recommended to be published after the following questions are clarified.
For oxidation study with weight gain experiment, such as Fig. 1, one of the major problem is the oxidation scales spalling, which make the weight loses in the measurement. This need authors to clarify.
Fig. 4, which is the AFT surface for the oxidation, does not provide useful information for the paper.
For the cross-section of oxide scale, such as Fig.5, authors should provide SEM image figure to give a full view of the oxide microstructures from the base-metal to the surface layer. Current TEM images only provide very localized information.
Current conclusion is not conclusive, very hand-wavy. For example, sentence “The HRSTEM-HAADF imaging revealed atom columns of the matrix and Ni3Nb particles” does not provide any useful information to reader. The precipitant in the alloy, should be results of annealing, not results of oxidation.
Author Response
Dear Reviewer,
Please find in the attached file response to your and other Reviewers' comments. As some of the questions may overlap I would like to provide the reply as one document.
best regards,
Sebastian

Reviewer 2 Report
Comments and Suggestions for Authors
Title: „Multiscale characterisation of an oxide scale formed on the creep-resistant ATI 718Plus superalloy during high temperature oxidation”, by Adam Kruk and coworkers.
Comments for the authors
This paper is focused on multiscale 2D and 3D characterization of the scale and the layer beneath formed on the ATI 718Plus superalloy during oxidation at 850 °C up to 4 000 hours in dry and wet air, and characterized using specific methods (SEM, TEM, STEM, XRD, AFM and EDXS). The authors show the difference between the structure, chemistry and three-dimensional phase distribution of the oxide scales and near-surface areas developed in the dry / wet air, and the importance of oxidation environment on the structure of the ATI 718Plus.
The introduction part explains the impact of water vapor on corrosion process, associated with the formation of volatile chromium oxyhydroxides, causing the protective properties of the Cr2O3 scale to deteriorate, reducing the maximum operating temperature and useful life of the alloys. Also, this study presents the main characteristics and the importance of ATI 718Plus in more aggressive environments, i.e. those containing water vapor. Also, this study presents the main characteristics and the importance of ATI 718Plus in more aggressive environments, and shows the characterization details of the 718Plus superalloy after oxidation at 850 °C up to 4000 h in two different media- wet 80 air (10 vol.% of H2O) and dry air.
The experimental part describes the chemical composition (Ni-18Cr-9.7Fe- 87 9.2Co-5.5(Nb+Ta)-2.7Mo-1W-1.5Al-0.7Ti-0.02C (wt%)), manufacturing (vacuum induction melting (VIM) and vacuum arc re-melting (VAR) followed by rotary forging and ring rolling), and thermal treatment (968 °C for 1 h, oil quenching, followed by two-step ageing at 788 °C for 8 h, furnace-cooled, and 704 ºC for 8h, air-cooled). The ATI 718Plus samples were oxidation at 850°C, using two different gas atmospheres: dry air (bubbled through deionized water kept at 45 °C) and humidified air with a 10 vol.% H2O content.
The characterization part is presented in detail, the ATI 718Plus samples were intensively studied by various characterization methods, the results indicating the behavior of this alloy in different conditions (dry and wet air), oxidation time (between 120h up to 4000h), temperature (850°C) and the processing conditions on the influence on mass loss, phase composition, topographic information, the oxidation environment on the complexity of the oxidized 718Plus structure, etc.
I consider that the results are good presented and corralled with the figures for each processing samples.
The conclusions are presented briefly.
Author Response
Dear Reviewer,
We are grateful to the referee for the kind opinion. Following the Reviewer's comment we enhanced the conclusions.
best regards,
Sebastian
Reviewer 3 Report
Introduction:
Lines 44-50: it is not clear if, in this part of the introduction, the authors are referring only to the ATI 718Plus alloy or it is a more general discourse. In fact, the authors start by considering only the Cr2O3 scale, without mentioning the reference alloy for which the reasoning applies, and then they return to talking about metal alloys in a general way.
Material and Methods:
Lines 101-103: “This temperature is higher than the typical operating temperature of the 718Plus, and was chosen to accelerate the oxidation processes and the microstructural changes and thus simulate the long-term operation of this alloy at lower temperatures”. Is this an acceptable hypothesis? is there any previous research paper showing that increasing the operating temperature simulates long-term operation at lower temperatures? We would need data to confirm that this hypothesis is verified and therefore acceptable for this type of alloy.
Results and discussion:
Line 166: why was it chosen to report in the graph in some cases the real values and in others the average of three measurements? If the average of several values is reported, their standard deviation must also be entered in the graph.
Line 209: “chemical composition analysis showed that these crystallites contained titanium”. Could this result be shown within the manuscript?
Figure 3: at what magnification were the enlarged frames performed?
Figure 3: the obtained results were only described and not discussed.
All the “results and discussion” section is very rich in the description of the reported images but it is very lacking in discussion and interpretation of the results obtained by the authors. The only hints of interpretation of the observed phenomena are in fact referred to other researchers' works.
It would also be useful to try to explain what is the main novelty of this work and how it can be important for the research field in which the authors are working. What is the most important application of the results obtained?
Author Response

(The authors gave the same response as above.)

Round 2
Reviewer 1 Report
Authors answer my previous questions, the paper now is OK for the journal.
Author Response
Dear Reviewer,
Thank you for your time.
best regards,
Sebastian Lech
Reviewer 3 Report
R#3: Material and Methods:
Lines 101-103: “This temperature is higher than the typical operating temperature of the 718Plus, and was chosen to accelerate the oxidation processes and the microstructural changes and thus simulate the long-term operation of this alloy at lower temperatures”. Is this an acceptable hypothesis? is there any previous research paper showing that increasing the operating temperature simulates long-term operation at lower temperatures? We would need data to confirm that this hypothesis is verified and therefore acceptable for this type of alloy.
Answer: It is common practice to increase the temperature to simulate the microstructural changes during long-term exposure. Nickel-base superalloys are designed to operate for even 100.000 or 200.000 hours at elevated temperatures, so it would take 10-20 years to see what are the exact microstructural changes of the superalloy. Larson-Miller parameter allows to calculate the increase of temperature needed to shorten the exposure time and is commonly used for such calculation. Moreover, such practice was used also by several other research groups and, among others, was published by B. Pint et al. (Ref. [4] given below) from Oak Ridge National Lab, US, who investigated oxidized 718 and 718Plus superalloys and clearly stated that (citation): "Exposures were conducted at 800 ºC to simulate long-term exposure at lower temperatures".
Furthermore, prior to the oxidation test, we have performed thermodynamic calculations of the equilibrium phase composition of the 718Plus superalloy via Thermo-Calc software. Below, please find the phase stability diagram as a function of temperature. Three main phases considered for this superalloy were γ, γ’ and δ phases. All three phases are stable to temperature above 850 °C. Thus, the oxidation test at this temperature have similar effect on the phase composition of this superalloy as oxidation at lower temperature. The main difference is the quantity of phases.
[4]. B.A. Pint, S. Dryepondt, K.A. Unocic, 7th Int. Symposium on Superalloy 718 and Derivatives, E. Ott et al. (eds), TMS 2010, p. 861.
Re-answer: Please add something in the manuscript that refers to this explanation provided by the authors.
R#3: Results and discussion:
R#3: Line 166: why was it chosen to report in the graph in some cases the real values and in others the
average of three measurements? If the average of several values is reported, their standard deviation must
also be entered in the graph.
Answer: We added the standard deviations for the oxidation times, where the value was an average of
several measurements.
Re-answer: I couldn't notice any standard deviation shown in the graph in figure 1.
Author Response
Dear Reviewer,
Thank you for the time taken to evaluate the manuscript.
We added the reference [18] to an appropriate sentence describing the method.
We have changed Figure 1 to the correct one. Furthermore, Figure 10 was replaced with a one of higher quality.
Best regards,
Sebastian Lech